

# Identification of hub genes and transcription factor-miRNA-mRNA pathways in mice and human renal ischemia-reperfusion injury

Peng Ke[1], Lin Qian[1], Yi Zhou[2], Liu Feng[1], Zhentao Zhang[2], Chengjie Zheng[1], Mengnan Chen[1], Xinlei Huang[1] and Xiaodan Wu[1]

[1] Department of Anesthesiology, Shengli Clinical Medical College, Fujian Medical University, Fuzhou, Fujian, China
[2] Institute of Pharmaceutics, College of Pharmaceutical Sciences, Zhejiang University, Hangzhou, Zhejiang, China

## ABSTRACT

**Background**. Renal ischemia-reperfusion injury (IRI) is a disease with high incidence rate in kidney related surgery. Micro RNA (miRNA) and transcription factors (TFs) are widely involved in the process of renal IRI through regulation of their target genes. However, the regulatory relationships and functional roles of TFs, miRNAs and mRNAs in the progression of renal IRI are insufficiently understood. The present study aimed to clarify the underlying mechanism of regulatory relationships in renal IRI.

**Methods**. Six gene expression profiles were downloaded from Gene Expression Omnibus (GEO). Differently expressed genes (DEGs) and differently expressed miRNAs (DEMs) were identified through RRA integrated analysis of mRNA datasets (GSE39548, GSE87025, GSE52004, GSE71647, and GSE131288) and miRNA datasets (GSE29495). miRDB and TransmiR v2.0 database were applied to predict target genes of miRNA and TFs, respectively. DEGs were applied for Gene Ontology (GO) and Kyoto Encyclopedia of Genes and Genomes (KEGG) analysis, followed with construction of protein-protein interaction (PPI) network. Then, the TF-miRNA-mRNA network was constructed. Correlation coefficient and ROC analysis were used to verify regulatory relationship between genes and their diagnostic value in GSE52004. Furthermore, in independent mouse RNA-seq datasets GSE98622, human RNA-seq GSE134386 and *in vitro*, the expression of hub genes and genes from the network were observed and correlation coefficient and ROC analysis were validated.

**Results**. A total of 21 DEMs and 187 DEGs were identified in renal IRI group compared to control group. The results of PPI analysis showed 15 hub genes. The TF-miRNA-mRNA regulatory network was constructed and several important pathways were identified and further verified, including Junb-miR-223-Ranbp3l, Cebpb-miR-223-Ranbp3l, Cebpb-miR-21-Ranbp3l and Cebpb-miR-181b-Bsnd. Four regulatory loops were identified, including Fosl2-miR-155, Fosl2-miR-146a, Cebpb-miR-155 and Mafk-miR-25. The hub genes and genes in the network showed good diagnostic value in mice and human.

Corresponding author
Xiaodan Wu, wxiaodan@sina.com

**Conclusions**. In this study, we found 15 hub genes and several TF-miRNA-mRNA pathways, which are helpful for understanding the molecular and regulatory mechanisms in renal IRI. Junb-miR-223-Ranbp3l, Cebpb-miR-223-Ranbp3l, Cebpb-miR-21-Ranbp3l and Cebpb-miR-181b-Bsnd were the most important pathways, while Spp1, Fos, Timp1, Tnc, Fosl2 and Junb were the most important hub genes. Fosl2-miR-155, Fosl2-miR-146a, Cebpb-miR-155 and Mafk-miR-25 might be the negative feedback loops in renal IRI.

## INTRODUCTION

Ischemia-reperfusion injury (IRI) is one of the main causes of acute kidney injury. Every year, one in every 2,000 people died of acute kidney injury caused by IRI (*Ali et al., 2007*). Renal IRI usually occurs after kidney transplantation, causing renal tissue damage and fibrosis (*Menke et al., 2014*), which plays adverse effects on the function of kidney and long-term prognosis of patients (*Tilney & Guttmann, 1997*). So far, there has no effective treatment for renal IRI.

Transcription factors (TFs) are proteins that can bind to target DNA and regulate the transcription level of target genes (*Hughes, 2011*). Many studies reported that TFs take part in the biological process of renal IRI. Previous studies found that in the early stage of renal injury, over expression of nuclear factor erythroid-2 related factor 2 (Nrf2) could delay the process of renal tubular injury by inducing antioxidant enzymes and nicotinamide adenine dinucleotide phosphate (NADPH) (*Nezu et al., 2017*). Other studies have found that transcriptional repressor GATA binding 1 (Trps1) is an essential TF for the formation of epithelial cells in the process of renal development, and the expression of Trps1 is positively correlated with the recovery of kidney after IRI. This suggests that Trps1 is a potential target for the treatment of renal ischemia-reperfusion injury (*Yang et al., 2017*). Recent studies have found that nicotiflorin can reduce the apoptosis of HK-2 cells by binding with activating transcription factor 3 (Atf3), delaying the process of renal IRI (*Wang et al., 2021*). In general, a considerable number of studies have shown that TFs are important intermediate molecules in renal IRI.

MiRNA, a kind of endogenous non coding RNA with a length of about 22 nucleotides, can bind to a variety of target mRNA and promote the degradation of target mRNA, inhibiting the expression of target gene and participating in the regulation of a variety of biological functions (*Ambros, 2004*). Previous studies showed that the expression of several miRNAs changed during renal IRI, including miR-21, miR-20a, miR-146a, miR-199a-3p, miR-214, miR-192, miR-187, miR-805 and miR-194 (*Godwin et al., 2010*). Other studies have found that miR-17-5p can directly inhibit the expression of death receptor 6 to reduce apoptosis and delay the process of renal IRI (*Hao et al., 2017*).

TF-miRNA-mRNA pathway is a kind of regulatory relationship and is crucial in the regulatory mechanisms in renal IRI. Recent years, studies have been reported that TFs regulate the expression of target genes through miRNAs in disease. However, the regulatory relationship of TF-miRNA-mRNA network in renal IRI is still unclear. In this study, we integrated analysis of 6 datasets from GEO database to provide a comprehensive understanding of renal IRI. The target genes and TFs of DEMs were predicted through online databases. Next, functional enrichment analysis of DEGs were applied and the hub genes in the PPI network were identified, followed with construction of TF-miRNA-mRNA network associated with renal IRI. Finally, the regulatory relationship and diagnostic value of genes were verified, which help to clarify the mechanisms and provide potential targets for renal IRI.

## MATERIALS & METHODS

### Collection of datasets

The mRNA expression profiles (GSE39548 (*Correa-Costa et al., 2012*), GSE52004 (*Liu et al., 2014*), GSE71647 (*Nezu, Suzuki & Yamamoto, 2017*), GSE87025 (*Markó et al., 2016*) and GSE131288 (*Aufhauser et al., 2021*)) and the miRNA expression profile (GSE29495 (*Shapiro et al., 2011*)) datasets were downloaded from the GEO database (http://www.ncbi.nlm.nih.gov/geo/) (*Barrett et al., 2013*). All of the datasets included renal IRI groups and control groups. Each group contained no less than 8 samples. Datasets were all available for download from GEO database. Flowcharts of the study was shown in 'Fig. 1'.

### DEGs and DEMs of renal IRI

The mRNA datasets were analyzed by NetworkAnalyst (*Zhou et al., 2019a*) and normalized before further analysis, followed with integrated analysis by "RobustRankAggreg" (RRA) package to define RRA DEGs in 5 mRNA datasets. The miRNA expression profile GSE29495 was analyzed to obtain the DEMs. The screening criteria for DEGs and DEMs were $|\log_2 FC| \geq 1$ and FDR < 0.05.

### DEGs enrichment analysis and construction of PPI network

Gene ontology (GO) and Kyoto Encyclopedia of Gene and Genome (KEGG) pathways of DEGs were analyzed by Metascape (*Zhou et al., 2019b*). DEGs list was submitted and analyzed as M. musculus. The PPI network and hub genes could be obtained quickly after clicking on "Express Analysis". Hub genes were defined according to the Molecular Complex Detection (MCODE) algorithm, usually with degree cutoff ≥3, K-core ≥4 and other parameters set as default values to identify densely connected network components, which were reported to have a higher probability to participate in biological regulation and disease status (*Bader & Hogue, 2003*).

### Prediction of differently expressed target genes and TFs of miRNA

The target genes of DEMs were predicted by online database miRDB (*Chen & Wang, 2020*), which was based on thousands of miRNA-target interactions from high-throughput

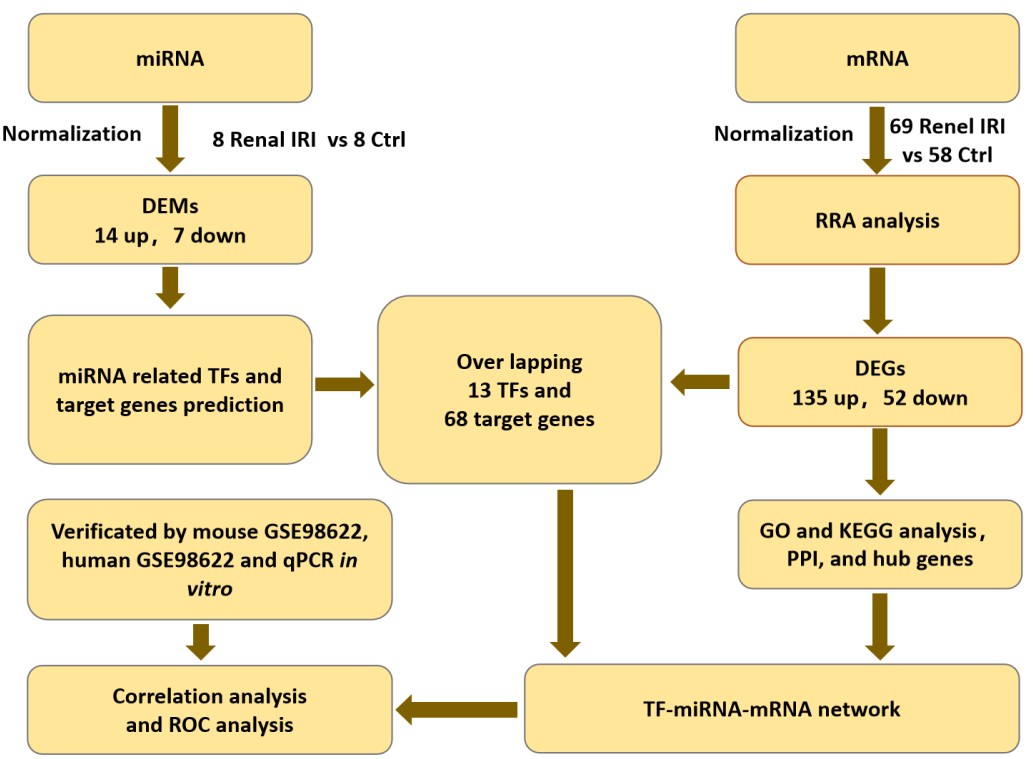

**Figure 1 Flowcharts of the study for analyzing TFs-miRNA-mRNA pathways network.** IRI, ischemia reperfusion injury; DEGs, differentially expressed genes; RRA, RobustRankAggreg; GO, gene ontology; KEGG, Kyoto Encyclopedia of Genes and Genomes; DEMs, differentially expressed miRNAs; TF, transcription factor; PPI, protein-protein interaction.

sequencing experiments and machine learning methods. The differently expressed target genes were obtained by overlapping DEGs and target genes of DEMs. The TransmiR v2.0 database (*Tong et al., 2019*), a database contains 1,785,998 TF-miRNA regulations derived from ChIP-seq evidence, was applied to predict TFs of DEMs for further overlapping with DEGs to obtain the differently expressed TFs. Therefore, differently expressed target genes and TFs were obtained.

## Construction of TF-miRNA-mRNA network

TF-miRNA-mRNA network was constructed based on differently expressed TFs, DEMs and target genes. Nodes that did not participate in the TF-miRNA-mRNA pathways were excluded. Then several important regulatory pathways were identified.

## Diagnostic value of genes and correlation coefficient between genes

In order to further verify the regulatory relationship between genes in the TF-miRNA-mRNA network, correlation analysis was used in R software. Generally, $|r| > 0.3$ indicates a correlation between TF and target gene. Besides, receiver operating characteristic curve (ROC curve) was applied in GraphPad Prism 7.0 to verify the diagnostic value of genes from the network for renal IRI. According to constructing of ROC model and calculating the area under the curve (AUC), it can better evaluate whether hub genes could be used

as biomarkers with high sensitivity and specificity in IRI model. Usually, AUC > 0.9 is considered to have good fitting effect in the model, which indicates that this gene has a good value in the diagnosis of diseases. GSE52004 datasets were set as the training data.

## Verification of the renal IRI associated hub genes and their diagnostic value through the analysis of RNA-seq data from mice and human

To verify the reliability of the experimental results, another independent RNA-seq datasets, GSE98622, was used to validate the expression of hub genes and their diagnostic value. RNA-seq data have the characteristics of more accurate, which makes our results more credible. GSE98622 was downloaded from GEO database and analyzed by NetworkAnalyst. 13 samples were excluded for quality control. Finally, 36 samples were included for further correlation analysis and ROC analysis. To verify the clinical value of the results, another human RNA-seq datasets GSE134386 were also applied to verify the expression of genes. GSE134386 contains nine expanded criteria donor samples and 8 living donor samples. The former kidneys experienced longer ischemia time in comparison with the latter ones. Finally, The Human Protein Atlas, available from http://www.proteinatlas.org, was applied to explore the expression of two target genes (Thul et al., 2017).

## The culturing of HK-2 cells and HR model

Human kidney 2 (HK-2), a proximal convoluted tubular epithelial cell, derived from normal kidney and was purchased from Procell Biotechnology Co. Ltd. (Procell). Cells were cultured in DMEM/F12 medium (BI) contained 10% fetal bovine serum (Giboco) and 1% penicillin streptomycin (Hyclone). The establishment of HR (hypoxia-reoxygenation) model was as follow (Wu et al., 2016). Hypoxia (1% oxygen) was performed in hypoxia chamber for 24 h after the medium was replaced with serum-free DMEM/F12 medium, and then incubated under normoxic conditions for 12 h. To determine the effect of Cebpb on Bsnd and Ranbp3l, Cebpb knockdown by siRNA for 12 h before HR.

## Real-time fluorescence quantitative PCR

Total RNA was extracted from HK-2 cells using EasySpin Plus Cell/Tissue RNA Rapid Extraction Kit (Adlai). PCR primers were designed and synthesized (File S9). RT-qPCR was performed on Lightcycle480 (Roche) using the SYBR Premixex Taq II (2X) (TAKARA). Fusion curves were performed after RT-qPCR to validate individual transcripts. All the reactions were repeated three times and the data were analyzed using the 2-$\Delta$ $\Delta$CT method. GAPDH was used as an internal control. The experiment was repeated three times independently.

## Statistical analysis

Statistical analysis was performed by using NetworkAnalyst and R software. Numerical data were presented as the mean $\pm$ standard deviation. DEGs and DEMs were calculated in NetworkAnalyst and defined according to the criteria of $|\log_2 FC| \geq 1$ and FDR $\leq 0.05$.

**Table 1  Details of the included microarray datasets.**

| GSE ID | Samples (IRI *vs* Ctrl) | Platform | Year | Type |
|---|---|---|---|---|
| GSE39548 | 12 *vs* 8 | GPL7202 | 2012 | mRNA |
| GSE52004 | 26 *vs* 31 | GPL6246 | 2014 | mRNA |
| GSE71647 | 4 *vs* 4 | GPL7202 | 2016 | mRNA |
| GSE87024 | 21 *vs* 9 | GPL6887 | 2016 | mRNA |
| GSE131288 | 6 *vs* 6 | GPL16570 | 2019 | mRNA |
| GSE29495 | 8 *vs* 8 | GPL13642 | 2011 | miRNA |

# RESULTS

## Expression profiles of mRNA and miRNA

The mRNA expression profiles were from five datasets, including GSE39548, GSE52004, GSE71647, GSE87025 and GSE131288. The miRNA expression profile was from GSE29459. There were 127 samples in mRNA expression profiles, including 69 renal IRI samples and 58 control samples. There were 16 samples in miRNA expression profile, including 8 renal IRI samples and 8 control samples (Table 1). The raw measurements were provided in the 'File S1'.

## Identification of DEMs and DEGs

The data distribution of each datasets were uniform after normalization. Boxplot and PCA showed that the qualified data can be used for further analysis (File S2). Compared with the control groups, 21 DEMs were identified in renal IRI groups, including 14 up-regulated miRNAs and 7 down-regulated miRNAs (Fig. 2A). After obtaining the DEGs of each mRNA gene expression profiles, 187 RRA DEGs were obtained by RRA integrated analysis, including 135 up-regulated genes and 52 down-regulated genes (File S3). The top ten up-regulated genes and the top ten down-regulated genes were shown in 'Fig. 2B'.

## GO and KEGG pathway analysis on DEGs and PPI network

GO and KEGG analysis of RRA DEGs were performed in Metascape. In the GO and KEGG analysis of DEGs, positive regulation of cell migration and TNF signaling pathway were significantly enriched. Interestingly, some GO items related to transcription factors had also been significantly enriched, including regulation of transcription from RNA polymerase âĔą promoter in response to stress and transmembrane receptor protein tyrosine kinase signaling pathway (Fig. 3A). All of the enrichment items were closely related, indicating that the pathogenesis of the renal IRI is complex (Fig. 3B). The PPI network showed 3 important MCODE modules (Fig. 3C), including 15 hub genes (Fig. 3D). Enrichment analysis on hub genes also showed that TNF signaling pathway was significantly enriched (File S4).

## Prediction of TFs and miRNA target genes

274 TFs were predicted by TransmiR v2.0 database based on miRNA-target interactions from high-throughput sequencing experiments (File S5), of which 13 differently expressed TFs were obtained by overlapping with DEGs (Fig. 4A). Based on this, regulatory network

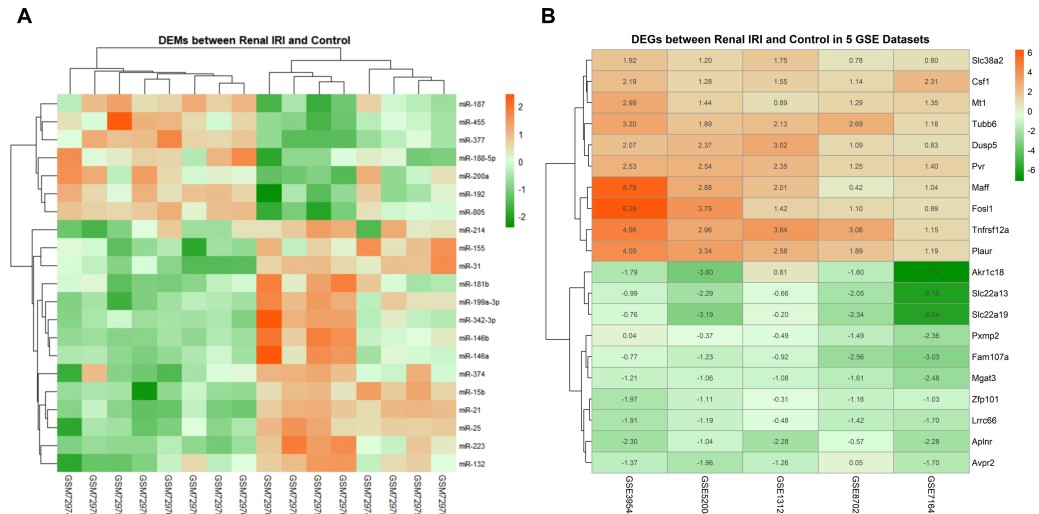

**Figure 2** **Heatmap of DEMs and DEGs expression profiles between renal IRI mice and controls.** (A) The heatmap showed 21 DEMs between renal tissues from renal IRI mice and controls. Each row represents different miRNA, and each column represents different sample from GSE29495. The samples on the right side of the cluster were renal IRI group. (B) Heatmap of the top 20 down and up-regulated DEGs from renal IRI mice in the result of RRA analysis. Each row represents different DEG, and each column represents different gene expression profile, indicating the consistency of DEGs in different gene expression profile. Red in the heatmap indicates high expression of miRNA or genes, while green represents low expression.

of TF-miRNA was constructed and showed 15 miRNAs regulated by 13 TFs 3 down-regulated miRNAs, including miR-455, miR-192 and miR-188-5p, were inconsistent with the regulation of TF activate miRNA (File S6). The results of miRDB showed that 6967 target genes were obtained (File S7), of which 68 genes were consistent with the DEGs (Fig. 4B). Among them, six target genes were TFs and marked in red. Therefore, regulatory network of miRNA-mRNA was constructed, which contains 12 miRNAs and 17 down-regulated targets (File S6).

## Construction of TF-miRNA-mRNA network

The TF-miRNA-mRNA network consists of 11 transcription factors (Egr1 and Nfil3 were deleted because they did not participate in the network), 10 miRNAs (miR-132 and miR-146a were deleted because they had no corresponding down-regulated target genes) and 17 target genes (Fig. 4C). Among them, Junb, Fos, Fosl1 and Fosl2 were hub genes, which showed their important roles in the network. In addition, outside of this network, we found four regulatory loops, including Fosl2-miR-155-Fosl2, Fosl2-miR-146a-Fosl2, Cebpb-miR-155-Cebpb and Mafk-miR-25-Mafk (Table 2), which indicated that TFs might maintain itself at a stable level through their target miRNAs.

## Diagnostic value of genes and Correlation coefficient between genes

In the TF-miRNA-mRNA network, we constructed a correlation analysis to show the correlation between various genes. The regulatory relationship we concerned were framed in red and blue. The red frames indicate that the correlation is satisfied. Atf4, Fosl1, Junb,

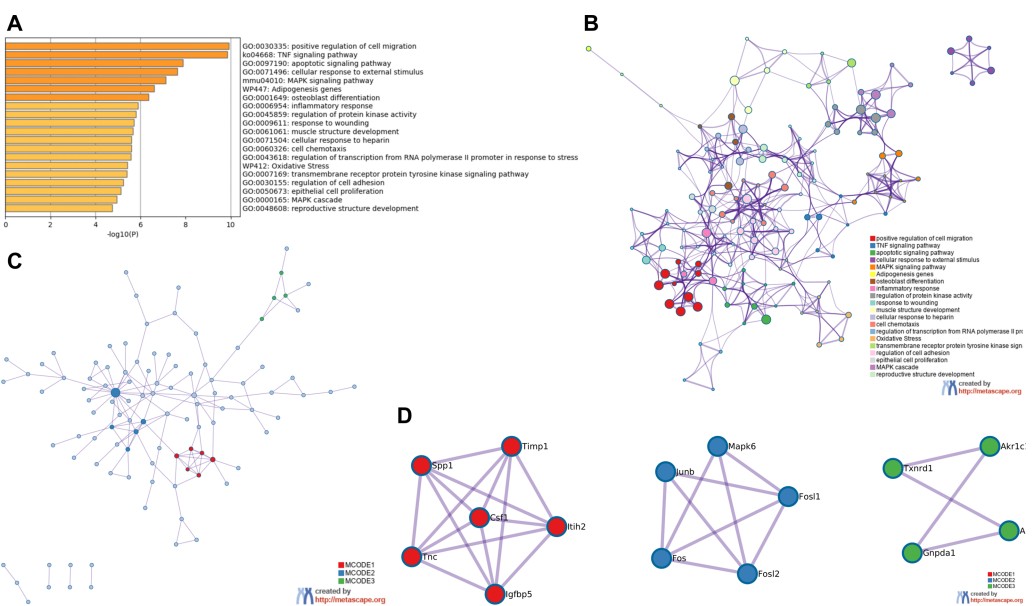

**Figure 3 Functional enrichment analysis of RRA DEGs and construction of PPI network.** (A) Bar graph for GO and KEGG enrichment analysis of RRA DEGs, one per cluster, using $p < 0.05$ to represent statistical significance. (B) Enrichment network clusters of RRA DEGs showing the connection and similarities between enriched terms. (C) The PPI network of DEGs associated with renal IRI. The red, blue and green modules showing their close connection in the network. (D) Modules in MCODE analysis were considered as hub genes and were marked in red, blue and green.

Fosl2, Egr2 and Cebpb were correlated with target genes ($|r| > 0.3$, $p < 0.05$) (Fig. 5A). Among them, Junb, Fosl1 and Fosl2 were hub genes, which showed their important roles in the network again. ROC curve showed that TFs and target genes from the network had a good accuracy in distinguishing renal IRI and control mice as well as the hub genes (Fig. 5B).

## Verification of the renal IRI associated hub genes and diagnostic value of target genes through the analysis of RNA-seq data from mice and human

RNA-seq data GSE98622 was performed to verify the expression levels of 15 hub genes and genes from the network. As shown in the analysis results, Timp1, Spp1, Csf1, Tnc, Junb, Fosl1, Fos, Fosl2 and Akr1b8 were up-regulated genes in renal IRI. While Itih2, Igfbp5 and Akr1c18 were down-regulated in renal IRI. The expression of Mapk6, Txnrd1 and Gnpda1 was not statistically significant between renal IRI and control group. The expression of genes in regulatory network were also verified in RNA-seq analysis. TFs, including Junb, Fos, Fosl1, Fosl2 and Egr2, were up-regulated in renal IRI. Although the expression of Cebpb was less than one, the difference was statistically significant. The change of Atf4 was not significant. Cyp7b1, A4gnt, Ranbp3l, Ass1, Acmsd, Bsnd and Galm were down-regulated target genes in renal IRI, while other target genes were not significant (Fig. 6A). The results of ROC analysis showed that most of hub genes shared good diagnostic value, although the expression of some genes changed less than one. Several genes from the
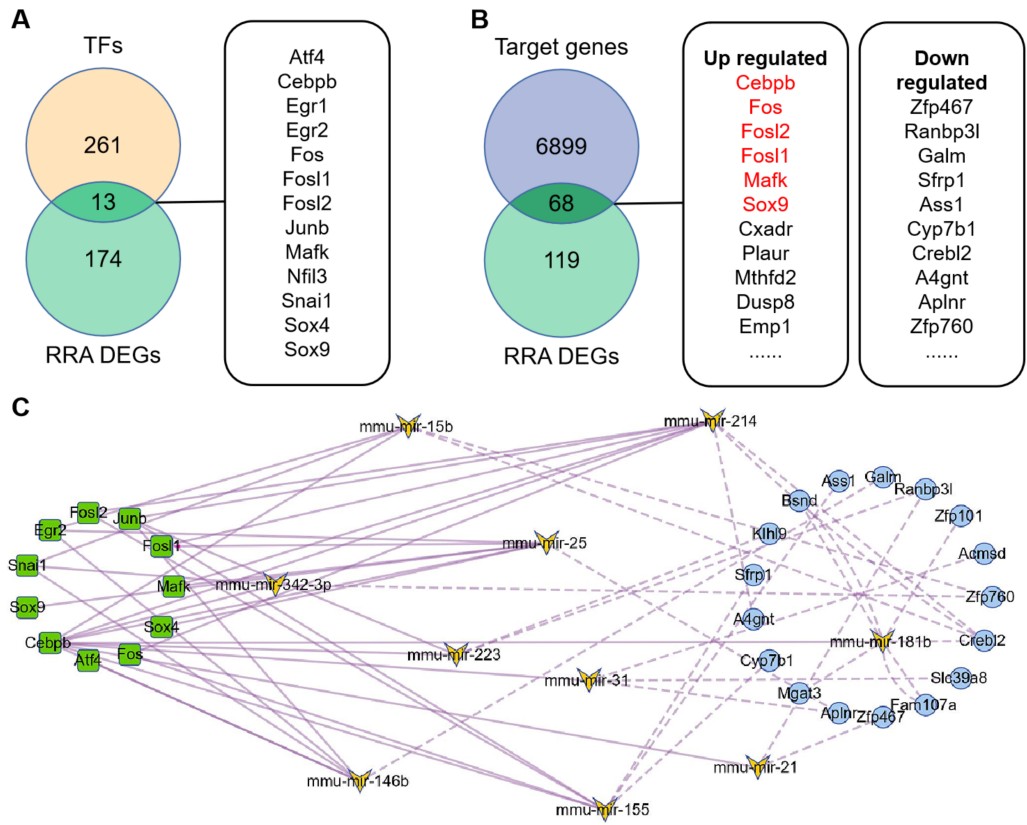

**Figure 4** **Construction of TFs-miRNA-mRNA pathways network associated with renal IRI.** (A) Venn diagram of TFs and RRA DEGs showed differentially expressed TFs. TFs were obtained according to TransmiR v2.0 database. (B) Venn diagram of RRA DEGs and target genes of DEMs showed differentially expressed target genes. Target genes of DEMs were predicted by miRDB and finally 68 genes were identified. The red target genes indicate that they are also transcription factors. (C) Construction of TF-miRNA-mRNA pathways network. The squares indicate differentially expressed TFs, the butterflies indicate differentially expressed miRNAs, the circles indicate differentially expressed target genes. The solid line indicates activation relationship between TFs and miRNAs. The dotted line indicates inhibition relationship between miRNAs and their target genes. 11 TFs, 10 miRNA and 17 target genes were incorporated into the network because they were accord with the regulatory relationship, and other genes were deleted because they were not accord with the regulatory relationship.

network that verified to be differently expressed, were all proved to share good diagnostic value again (Fig. 6C). Correlation analysis showed good correlation between differently expressed TFs and target genes from the network (Fig. 6B). Consistent with the previous results of GSE52004, TFs, including Fosl1, Junb, Fosl2, Egr2 and Cebpb, were negatively regulated with their corresponding target genes in GSE98622. However, there was no correlation between Atf4 and target genes. Interestingly, Fos showed a correlation with Ass1 and Crebl2, which was inconsistent with that in GSE52004. In human samples, hub genes, including SPP1, FOS, TIMP1, TNC, FOSL2 and JUNB, were verified to be up-regulated in severely damaged kidneys. In the regulatory network, EGR2 and FOS were significantly up-regulated. CEBPB, FOSL2 and JUNB were also up-regulated, with the

![PeerJ]

**Table 2  Details of four negative feedback loops in renal IRI.**

| TF | miRNA name | Binding site | SRAID | miRNA Sequence | Target Score |
|---|---|---|---|---|---|
| Fosl2 | mir-155 | chr16:84703195-84703369 (score = 647) | SRX187214 | UUAAUGCUAAUUGUGAUAGGGGU | 73 |
| Fosl2 | mir-146a | chr11:43377227-43377530 (score = 541) | SRX187215 | UGAGAACUGAAUUCCAUGGGUU | 51 |
| Mafk | mir-25 | chr5:138165696-138165895 (score = 1000) | SRX188829 | CAUUGCACUUGUCUCGGUCUGA | 71 |
| Cebpb | mir-155 | chr16:84712939-84713250 (score = 447) | SRX2901279 | UUAAUGCUAAUUGUGAUAGGGGU | 88 |

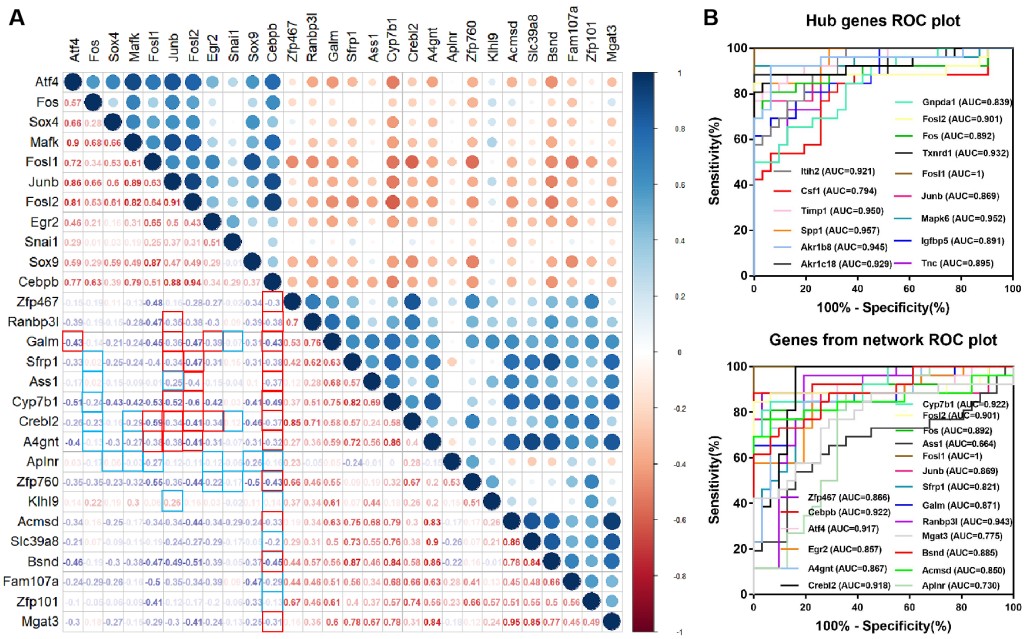

**Figure 5  Diagnostic value of hub genes and Correlation coefficient between TFs and target genes.** (A) Correlation analysis between differentially expressed TFs and differentially expressed target genes in the network. The frames marked in red and blue indicate regulatory relationship between TFs and targets. The red frame in the lower left corner indicates correlation, and the blue frame indicates no correlation. (B) Diagnostic value of hub genes and genes from the network in renal IRI in GSE52004.

fold change varied between 0.5 and 1. Expression of BSND and RANBP3L from network showed significantly changed, which was consistent with the observation in GSE98622 (Fig. 6D). The Human Protein Atlas showed that Bsnd and Ranbp3l were mostly expressed in normal human kidney tissues, especially Bsnd is uniquely up-regulated in the kidney, which suggest their importance in maintaining normal physiological function (Fig. 7A). The protein expression levels of Bsnd and Ranbp3l were consistent with their mRNA expression (File S8). Immunohistochemical staining showed that Bsnd and Ranbp3l were highly expressed in normal kidney tissues (Fig. 7B). The figures were downloaded from The Human Protein Atlas (https://www.proteinatlas.org/ENSG00000164188-RANBP3L/tissue and https://www.proteinatlas.org/ENSG00000162399-BSND/tissue).

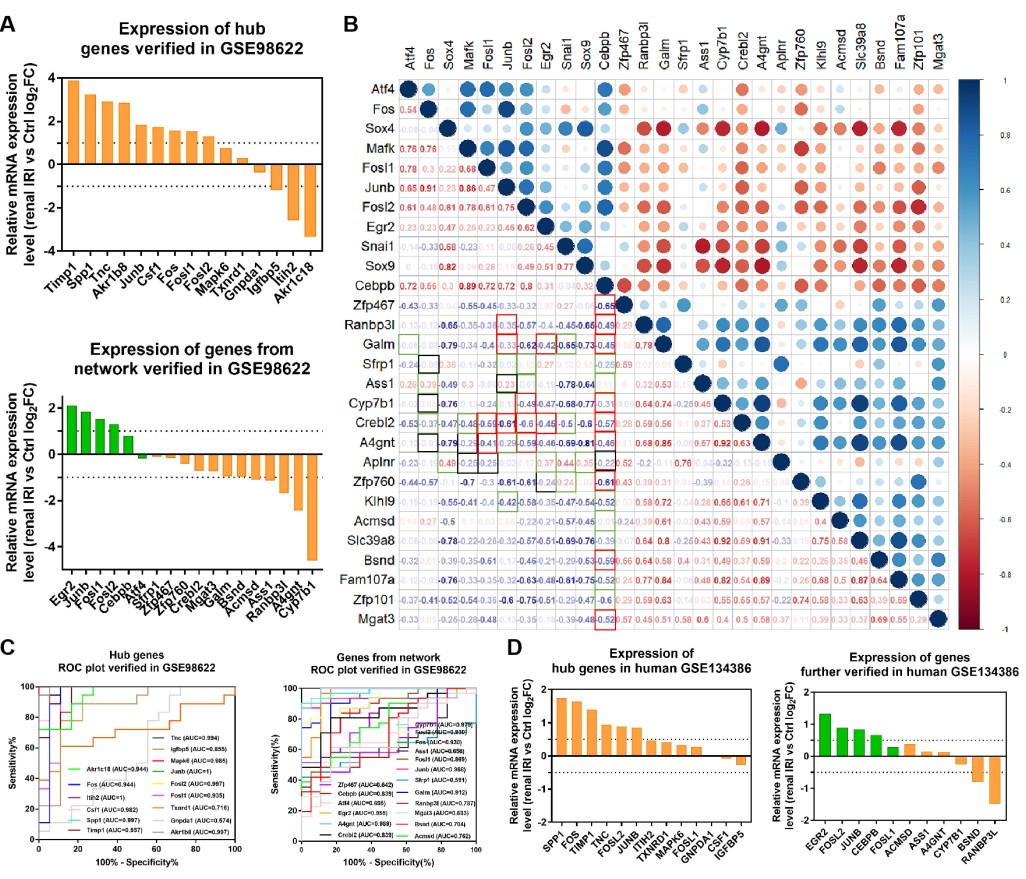

**Figure 6** Verification of the expression of genes and their diagnostic value, and correlation between the genes in mice RNA-seq datasets GSE98622 and human datasets GSE134386. (A) Relative mRNA expression of hub genes and genes from the network in GSE98622. TFs were marked in green, target genes were marked in orange. (B) Correlation analysis between TFs and target genes from the network in GSE98622. Black indicates irrelevant again; Red indicates relevant again; Green indicates that the two results were inconsistent. (C) Diagnostic value of hub genes and genes from the network in renal IRI in GSE98622. (D) Relative mRNA expression of hub genes and genes from the network in human, which were also differently expressed in mouse samples.

## Verification of the genes from the network *in vitro*

The expression of Fos, Egr2, Junb, Fosl2, Cebpb, Bsnd and Ranbp3l were verified in HK-2 cells *in vitro*. Fos, Egr2, Junb, Fosl2 and Cebpb were up-regulated in HR group, while Bsnd and Ranbp3l were down-regulated (Fig. 7C). After knockdown of Cebpb, the expression of Cebpb was down-regulated, while Bsnd and Ranbp3l were up-regulated (Fig. 7D).

## DISCUSSION

The incidence of renal IRI is increasing recent years. 15 to 20 min of ischemia is enough to cause kidney damage because of abundant blood supply and high metabolism of kidney (*Schrier & Wang, 2004*). Currently, lots of research found genetic alteration after renal IRI. Some achievements have been made in genes expression modification (*Tobisawa et al., 2017*) and molecular treatment (*Yip et al., 2015*). Previous microarray studies have

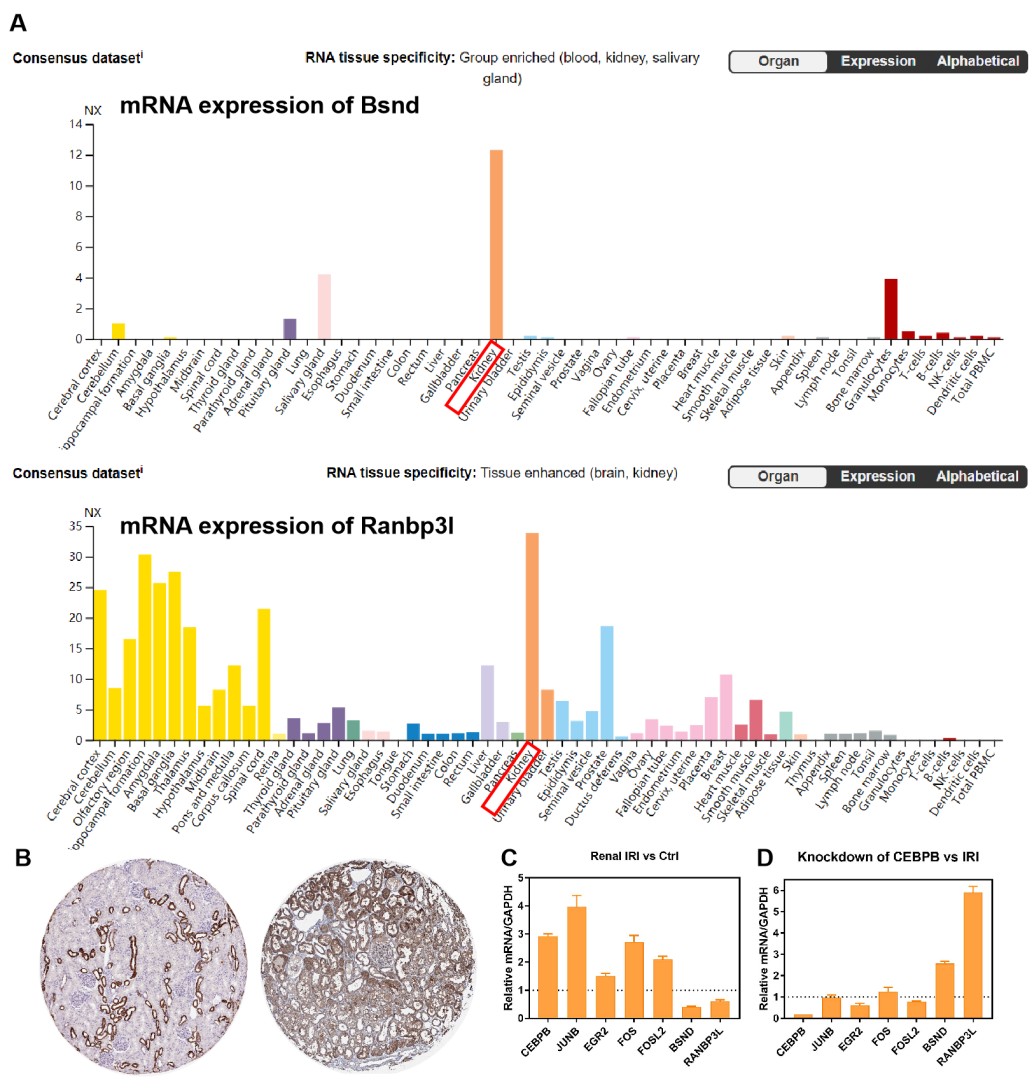

**Figure 7** **Verification of the expression of genes *in vitro*.** (A) The expression of Bsnd in normal kidney tissues. The orange modules show that the Bsnd and Ranbp3l are highly expressed in normal kidney. Downloaded from The Human Protein Atlas. (B) Immunohistochemical staining showed that Bsnd and Ranbp3l were highly expressed in normal kidney tissues. Left, Bsnd; right, Ranbp3l. Downloaded from The Human Protein Atlas. (C) The expression of genes in HK-2 HR model *in vitro*. (D) Bsnd and Ranbp3l were up-regulated after knockdown of Cebpb *in vitro*.

often focused on the role of a single molecule in renal IRI while ignoring the important regulatory relationship between genes. Regulation of miRNAs and TFs on target genes in renal IRI still remain unclear. In this study, five mRNA expression profiles were integrated to reduce interference factors, which was better than single gene expression profile analysis. 21 DEMs and 187 DEGs were identified. 15 hub genes were identified in renal IRI. Several differently expressed TFs and target genes were predicted based on over lapping with DEGs, followed with the construction of TF-miRNA-mRNA regulatory network. Finally, biological significance of these genes and the regulatory relationship between TFs and

mRNAs were verified by the independent RNA-seq datasets GSE98622, human datasets GSE134386 and *in vitro*.

Several DEMs were up-regulated in our result and have been reported in renal IRI. Some of the DEMs have shown to alleviate kidney injury. MiR-21 in exosomes inhibited programmed cell death protein 4 (PDCD4) and reduced apoptosis of tubular epithelial cell (*Du et al., 2020*). MiR-146b, targeting interleukin (IL)-1 receptor-associated kinase (IRAK1), could reduce sepsis-associated acute kidney (*Zhang et al., 2020*). Exosomes delivering miR-223 down-regulated NLR Family Pyrin Domain Containing 3 (NLRP3) (*Yuan et al., 2017*). Injection of exosomes carried with miR-199-3p decreased the expression of semaphorin 3A (Sema3A), while activated Akt and ERK signaling pathways (*Zhu et al., 2019*). Li Xirui et al. found that miR-146a from exosomes protect renal IRI in rats (*Li et al., 2020*). The up regulation of these miRNAs in our study might be caused by the protective effect of the body. Some of the DEMs have been demonstrated to exacerbate kidney injury. MiR-132 promote renal tubular apoptosis (*Yan et al., 2020*), which plays the same effect as miR-155 (*Wu et al., 2016*). MiRNA-214 represses mitofusin-2 (*Yan et al., 2020*) and disrupts mitochondrial oxidative phosphorylation to promote renal IRI (*Bai et al., 2019*). These results are consistent with the results of existing research, which strengthen the reliability of our analysis results.

Among the down-regulated DEMs, some miRNAs play aggravating roles in renal IRI. For example, MiR-377 aggravates renal IRI by inhibiting vascular endothelial growth factor (VEGF). Inhibition of miR-377 improves kidney injury (*Liu et al., 2019*). miR-188 was reported in ischemia-reperfusion injury of retinal ganglio cells (*Ge et al., 2020*). miR-31 was also up-regulated in cardiac IRI (*Wang et al., 2015*). MiR-200a is often up-regulated in the early stage of injury (*Sonoda et al., 2019*). Here, we found miR-200a was down-regulated in the study and need more study to explore it. Conversely, some miRNAs play protective roles in IRI. MiR-192 was down-regulated in renal IRI, which was consistent with our results (*Zhang et al., 2017*). MiR-187 inhibit apoptosis by targeting acetylcholinesterase (ACHE) in mouse podocyte (MPC-5) cells, which was consistent with miR-187 agomir injection in mice (*Yue et al., 2019*). MiR-455 was also associated with cerebral IRI (*Fan et al., 2021*).

Studies reported that miR-181b, miR-25, miR-342-3p, miR-374 and miR-15b were involved in IRI but not in renal IRI. Therefore, their roles in renal IRI require further exploration. Besides, miR-15b, miR-214, miR-25, miR-342-3p, miR-223, miR-31, miR-181b, miR-146b, miR-155 and miR-21 were in the TF-miRNA-mRNA network we built. In addition, miR-31 has never been reported in IRI.

Construction of PPI network is very commonly used in transcriptome analysis. 15 hub genes, including Timp1, Spp, Csf1, Itih2, Tnc, Igfbp5, Mapk6, Junb, Fosl1, Fos, Fosl2, Txnrd1, Gnpda1, Akrlc18 and Akrlb8, were identified according to the application of MCODE algorithm. A large number of studies have adopted the results of MCODE as hub genes for further analysis. These hub genes have higher probability to participate in biological regulation and correlated with renal IRI and could act as biomarkers, which might be applied to estimate the severity of IRI and verify effectiveness of treatments. Several genes has been reported in renal IRI, which indicate the accuracy of our analysis.

Timp1 is the marker of renal injury (*Deng et al., 2021*). Fos mRNA expression was rapidly and briefly up-regulated following renal ischemia (*Megyesi et al., 1995*). Spp1 mainly takes part in the process of osteoblast differentiation and inflammatory response (*Srirussamee et al., 2019*). Cebpb activates miR-16 to promote renal injury (*Chen et al., 2016*). Interestingly, 4 hub genes were in the TF-miRNA-mRNA network, including Junb, Fos, Fosl1 and Fosl2. While the role of Junb and Fosl2 in renal IRI remains unclear.

In the results of genes enrichment analysis, we found DEGs were significantly enriched in TNF and MAPK signaling pathway. All of them have been reported in renal IRI. The results of GO analysis showed that regulation of transcription from RNA polymerase π promoter in response to stress and transmembrane receptor protein tyrosine kinase signaling pathway were significantly enriched, indicating that DEGs in renal IRI were closely related to transcription factors. Therefore, TFs of DEMs were predicted in the study. A total of 13 differently expressed TFs were found, including Atf4, Cebpb, Egr1, Egr2, Fos, Fosl1, Fosl2, Junb, Mafk, Nfil3, Snai1, Sox4 and Sox9, suggesting their important roles in renal IRI. When constructing TF-miRNA, 2 TFs and 3 miRNAs were deleted due to their inconsistent expression regulation. 2 miRNAs and up-regulated targets were also deleted during the construction of miRNA-mRNA because of their negative regulatory relationship.

Finally, the TF-miRNA-mRNA network was constructed, which consisted of 11 TFs, 10 miRNAs and 17 target genes. All miRNAs in the network were up-regulated DEMs in renal IRI, which also surprised us. There might be three reasons. Firstly, almost all of the transcription factors we predicted activate their target miRNAs, indicating that target miRNAs should be up-regulated. Thus, 7 down-regulated miRNAs were no longer in our concern. Secondly, we did observe that TF inhibited miRNA, such as Cebpb inhibited miR-146a. However, the expression of miR-146a was actually up-regulated in IRI, which was contrary to their negative regulatory relationship. Thirdly, we deleted the DEMs or DEGs that are not involved in TF-miRNA-mRNA network and only showed the most important information. Correlation analysis showed that several regulatory pathways were correlated, which were consistent with our prediction. Unexpectedly, the expression of some other genes were correlated, possibly because they are regulated or affected by other more complex mechanisms. For example, some TFs might binding to their target mRNAs directly. Besides, ROC curves showed the diagnostic value of genes in the network. The role of ROC analysis verified that these genes correlated with renal IRI and could act as biomarkers or therapeutic targets, which might be applied to estimate the severity of IRI and verify effectiveness of treatments. In addition, we also found four regulatory loops, including Fosl2-miR-155-Fosl2, Fosl2-miR-146a-Fosl2, Cebpb-miR-155-Cebpb and Mafk-miR-25-Mafk. Regulatory loop has been reported in liver ischemia-reperfusion injury (*Pan et al., 2020*). Our results showed that transcription factors bind to the promoter of miRNA and induce miRNA express, so as to form a negative feedback loop with TF and maintain the stability of their expression level.

To verify our analysis results, RNA-seq data GSE98622 was applied for analyzing the expression of genes, ROC values and correlation analysis between genes, showing good support for our analysis results. Based on the results of GSE98622, regulatory relationships

between TFs and target genes were verified. Finally, to explore the value of these genes in clinical practice, expression of hub genes and genes from network were analyzed in human. To our excitement, the expression of 6 hub genes (SPP1, FOS, TIMP1, TNC, FOSL2 and JUNB) were significantly up-regulated in more severely damaged kidneys, which was highly consistent with the results in mice. The expression of 5 TFs (FOS, EGR2, JUNB, FOSL2 and CEBPB) and 2 target genes (BSND and RANBP3L) were also significantly changed, indicating their importance in human IRI. BSND and RANBP3L have never been reported in renal IRI. These results suggest that these genes have great research value in humans. Many genes have no significance in human samples, which may be caused by the differences of species. Based on the above results, we consider Junb-miR-223-Ranbp3l, Cebpb-miR-223-Ranbp3l, Cebpb-miR-21-Ranbp3l and Cebpb-miR-181b-Bsnd as the most important pathways, and Spp1, Fos, Timp1, Tnc, Fosl2, Junb, Egr2, Cebpb, Bsnd and Ranbp3l as the most important genes. Thus, we verified the expression of Fos, Egr2, Cebpb, Junb, Fosl2, Bsnd and Ranbp3l *in vitro*. Bsnd and Ranbp3l were mostly expressed in normal human kidney tissues, which suggest their importance in maintaining normal physiological function. Both of them were down-regulated after IRI, and were up-regulated after knockdown of Cebpb in our study, because the decrease of Cebpb leads to the down-regulated of its target miRNA (miR-223, miR-21 and miR-181b), the latter inhibit the expression of Bsnd and Ranbp3l. The expression of Cebpb and target miRNA are up-regulated during renal IRI, and can aggravate the damage, while Randp3l is down-regulated. Randp3l has been reported to play a protective role in renal carcinoma (*Chernyakov et al., 2021*). Therefore, these regulatory relationships are of great significance for further research, we would further confirm them experimentally in the future.

In this study, we have taken many measures to ensure the reliability of the study. For example, RRA algorithm was applied to integrated analysis on multiple datasets. All of the datasets were from the same species and the data were normalized before further analyzed. Two independent RNA-seq datasets and HK-2 cells were applied to verify our results. However, there are still some limitations. Firstly, DEGs analysis would inevitably filter out some important genes with small changes. Secondly, the model condition of renal IRI from different datasets are not exactly the same. Thirdly, we didn't explore the specific molecular functions of these hub genes in renal IRI. In addition, the construction of TF-miRNA-mRNA pathways network lost some hub genes not associated with DEMs, but it can also filter out many gene not involved in TF-miRNA-mRNA pathways regulation to make the analysis more convincing. Because of RNA-seq datasets contain no miRNA results, the verification of miRNAs in the pathways was indirectly. Due to the small sample size in human datasets, ROC analysis and correlation analysis are difficult to carry out. Further experimental studies are needed to prove the relationship between TF-miRNA-mRNA pathways and four negative feedback loops.

## CONCLUSIONS

In summary, we present a bioinformatics analysis of mRNA and miRNA expression and identified several hub genes, TF-miRNA-mRNA regulatory pathways and four negative

feedback loops that might play important roles in renal IRI, which provide potential therapeutic targets and deeper understanding of genetic mechanism for renal IRI.

### Funding
This work was supported by the Joint fund for research and development of high-level hospital (No. 2017LHJJ10). The funders had no role in study design, data collection and analysis, decision to publish, or preparation of the manuscript.

### Grant Disclosures
The following grant information was disclosed by the authors:
The Joint fund for research and development of high-level hospital: No. 2017LHJJ10.

### Competing Interests
The authors declare there are no competing interests.

### Author Contributions
- Peng Ke, Liu Feng and Xiaodan Wu conceived and designed the experiments, performed the experiments, analyzed the data, prepared figures and/or tables, authored or reviewed drafts of the paper, and approved the final draft.
- Lin Qian and Zhentao Zhang performed the experiments, analyzed the data, prepared figures and/or tables, authored or reviewed drafts of the paper, and approved the final draft.
- Yi Zhou performed the experiments, prepared figures and/or tables, authored or reviewed drafts of the paper, and approved the final draft.
- Chengjie Zheng, Mengnan Chen and Xinlei Huang performed the experiments, analyzed the data, authored or reviewed drafts of the paper, and approved the final draft.

### Data Availability
The raw measurements are available in the Supplemental Files.

### Supplemental Information
Supplemental information for this article can be found online at http://dx.doi.org/10.7717/peerj.12375#supplemental-information.

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
