# Peer review of "Identification of hub genes and transcription factor-miRNA-mRNA pathways in mice and human renal ischemia-reperfusion injury"

_PeerJ, doi:10.7717/peerj.12375_

## Round 0.1 · original submission · Major Revisions

The manuscript should be changed in accordance with the reviewers' suggestions. Some major points:

- Please change the title to better describe the study;

- Please clarify and re-think definitions for hub genes and loop;

- Add limitations of the study to the corresponding paragraph in the Discussion section.

Reviewer 1 ·

Basic reporting

This report uses published data sets to examine possible transcription factor-miRNA-mRNA interactions or roles in renal ischemia repercussion injury. Overall, the presentation of the data us within the standards of the literature.

Experimental design

Published data sets were analyzed using a variety of on-line tools to examine how IR alters gene expression and its regulation. The data analysis are straight forward.

Validity of the findings

The major issue with the paper is the lack of data showing experiment based independent validation of the differential expression patterns observed. No data validating the importance of identified hubs are presented.

Additional comments

I don't think this report contributes much. The authors should provide a more in-depth analysis to address whether the changes observed are IR specific.

Reviewer 2 ·

Basic reporting

There are not enough descriptions in figure legends in all figures. And some of them are not correct. Therefore, it’s very difficult to understand the results. The authors need to add more detailed descriptions in all figure legends.

Fig2A is not clear because the legends don’t have enough description. One sample per lane with data set number (at the bottom) is shown? One half is ISI samples and the other half controls in this heatmap? Legends say 32 DEMs but I see only 21 DEMs in this figure.

Fig.3C. In this study, RNA levels were examined. Therefore, protein-protein interaction doesn’t provide much information. It is better to search genes by initial genes in the pathways (as hub genes?).

Fig.3D. Because the details of MCODE module analysis is not clear, it is not sure if these genes are hub genes. Definition of “hub gene” should be described. This study may just pick the proteins which interact with many proteins. Fig.3A suggests more TNF related genes but the identified hub genes here don’t match with them and even the genes identified in Fig.2A. It may suggest that the critical genes are not regulated by miRNAs. It is better to change the strategy to identify critical genes. Again, it is better to search genes by initial genes (as hub genes) in the pathways.

Fig.4. Did they analyzed all DEMs or only increased DEMs in IRI cases? According to Fig.4C, all miRNAs are increased DEMs in IRI. If so, decreased DEMs should be examined in the same way.
It is difficult to understand why they are in loop. They are simply pathways. Does it mean negative feedback regulation of same family TF mediated by miRNAs? Are there any examples of “positive” feedback? Positive feedback loops mediated by miRNAs were reported in kidney diseases not only IRI.

Fig.5. Because it is not sure if these genes are hub genes, the study doesn’t sound meaningful.
Fig.5A. If those TFs upregulates miRNAs, targets of miRNAs should be down regulated (negatively correlated). But all of them (TFs and targets) are positively correlated. This is another unclear part of this study. Identified TFs and targets may be directly related and miRNAs may not be involved in the signaling.
Fig.5B is not clear, either. What is the meaning of “diagnostic” here? Is it related to IRI?

Fig.6. Why was the RNA-seq results used as confirmation? DEGs identified in this dataset (RNA-seq) may be analyzed like former analysis and get more targets? Again, same as Fig.5, these results may just show positive correlation of TFs and targets genes and miRNAs may not be involved (because no miRNA results are in this dataset (RNA-seq).

Experimental design

Definition of “hub gene” should be described. In this study, RNA levels were examined. Therefore, protein-protein interaction doesn’t provide much information. This study may just pick the proteins which interact with many proteins. It is better to change the strategy to identify critical genes. It is better to search genes by initial genes (as hub genes) in the pathways.

Methods
MCODE module analysis should be explained more.

Diagnostic value of genes and COR curve should be described more. Because Method is not clear, results are not so informative.

This study is based on mouse IRI model. Because human patient data (microarray or RNA-seq) should be available, comparison of the results the authors obtained with mouse data in this study with human data is encouraged.

The authors collected datasets of mRNA and miRNAs studied in mouse IRI models from publicly available databases and analyzed DEGs and DEMs and made connections from TF to miRNAs and mRNA (targets), and found some possible pathways involved in IRI. But all of them are correlative and not proved well by experiments. Testing some of the most important pathways by the experiments (in cultured cells) is encouraged.

Validity of the findings

As mentioned above, it is not sure if the genes the authors identified are hub genes. Reconsider the definition of hub gene and search for real hub genes (such as initial genes of the pathways but not based on protein-protein interaction).

Reconsider the definition of loop. They identified some signaling pathways but they are not in loop. If TF activate miRNA which targets the "same TF", it may be called loop. But miRNA targets TF but not exactly same TF.

In those pathways, TF activates miRNA which targets (downregulates) target. In this situation, TFs and targets should show negative correlation. But in this study, all of them showed positive correlation. It could be just direct regulation of targets by TFs and miRNAs may not be involved.

Additional comments

Title is misleading.
They identified pathways of TF-miRNA-mRNA but not TF-miRNA-mRNA. “Pathway” should be included.
They used mouse IRI models but not human patients. It is better to include “mouse models” of IRI.

---

## Round 0.2 · Major Revisions

Please carefully address the concerns raised by reviewer #2.

Reviewer 2 ·

Basic reporting

Revised manuscript has more descriptions and is easier to understand for readers. But because of that, more concerns have been raised up. I am not sure if continued transcription factor miRNA-mRNA pathways exist or not.

Several papers reported that miR-155 is upregulated in kidney injury and miR-155KO mice are protected. It is totally opposite of the authors’ conclusion.
Authors also discussed that miR-155 induces kidney injury in discussion. Such discrepancy should be discussed.
miR-192 and miR-200 family are upregulated in this study and also in previous reports of kidney injury. Those upregulated miRNAs should be discussed.

Many of Hub genes identified in this study and Fos/Jun family proteins are already known to be upregulated in IRI.

Overall, this manuscript doesn’t have much novelty but is just confirmation of known biomarkers identified by rescreening of publicly available RNA datasets (without any experiments using kidney cells). (confirming Fos, Fosl2, miR-155 and other RNAs as markers for IRI but no proof of regulation or pathways.)

Experimental design

The authors reanalyzed (correlation of the expression of DEM and DEG or TFs) the publicly available RNA datasets in human and mouse IRI (without any experiments using kidney cells).

Authors searched hub genes based on protein-protein interaction of predicted proteins from DEG in IRI and tested correlations of hub genes and IRI.

Validity of the findings

As also mentioned by authors, all of DEMs identified in this screening are down-regulated. It is quite surprising. Authors mentioned among 21 DEMs, only 7 miRNAs were up-regulated. But 7 out 21 is quite significant number. I don’t understand why they lost those 7 miRNAs in the analysis. TFs identified in this study are activators of transcription. But miRNAs DEMs in this study are down in IRI. It doesn’t make sense. Supplementary file 5 is not appropriate and not the evidence. Figures of correlation between TFs and DEMs (like TFs and targets) should be provided.

Authors used two databases, FunRich (http://funrich.org) and TransmiR v2.0 database. They need to explain how those databases identified TF for miRNAs. Simply expression or location of TF sites around miRNAs or experimentally proved?

As commented in the first review, this is a totally correlative study but no guarantee of regulation. Even if TF and DEM pair is found, that doesn’t mean TF regulates DEM. DEM may regulate TF identified in this study, too. It could also be just coincidence (no regulation for each other). Figures of correlation between targets and DEMs (like TFs and targets) should be provided.

Fig.5 & 6 just show positive correlation of TFs and target genes, but miRNAs may not be involved. Because same family TFs are both in TFs and targets, simply TFs increase target RNAs (same family). Again, it is difficult to understand why miRNAs are downregulated by those TFs (activating transcription).

Initial downregulation of miRNAs may regulate TFs (which could upregulate target mRNAs). It could be simply miRNA-TF (target) but not TF-miRNA-mRNA. Upregulated Fos and fosl2 and downregulated miR-155 can be markers for IRI. However I doubt if TF-miRNA- target pathway is real. It could be simply miRNA-target or TFs-targets (no miRNA involved).

Hub genes are identified by potential protein-protein interaction using RNA datasets (as translated proteins). Since the same family proteins are interacting each other, such proteins can be preferentially picked. It is difficult to be combined with microRNAs data although the expression of hub genes is correlated with IRI.

To make those concerns addressed, in vitro experiments are necessary, as also suggested by another reviewer. Especially Fos and JunB are positive TFs and upregulate gene expression. It is difficult to believe that Fos and Junb downregulate miRNAs. Again, figures of correlation between targets and DEMs should be provided. Fos or Junb may simply (directly) upregulate Fosl2 (without miR-155). Analyzing other RNA datasets doesn’t help at all.

I think experiments using cultured kidney cells should be done to solve this puzzle. Overexpression of Fos or Junb down (or maybe up) regulates miR-155 in cultured cells? miR-155 decreases Fosl2 and miR-155 inhibitor increases Fosl2 in in vitro culture experiments? Overexpression of hub genes changes the expression of those miRNAs and targets? Overexpression or inhibition of TFs or miRNAs changes the expression of hub genes?

Or (without in vitro experiments), tone down “hub genes and pathways” (should be excluded from title and abstract).
Authors should discuss at least four possibilities (not only the first one).
TF-miRNAs-Targets (very weak evidence)
miRNAs-TFs (can also upregulate targets)
miRNAs-Targets (can also be TFs)
TFs-Targets (without miRNAs)
(hub genes can be TFs or targets)

Therefore, the major conclusion and the title (pathways and hub genes) of this study are not supported by their results. This is just a reanalysis of publicly available RNA datasets confirming known facts (biomarkers of IRI).

Additional comments

It may be better to test (emphasize) interactions (or positive or negative correlation) of previously unknown DEM and DEG by experiments rather than confirming known DEM and DEG in IRI.

Based on the authors' current results, Title may be
"in silico confirmation of biomarker RNAs of human and mouse IRI by reanalyzing publicly available RNA datasets."

Typo in abstract
Fosl2 is correct but not "Fols2".

---

## Round 0.3 · accepted · Accept

The answers to the reviewer's comments are satisfactory.